# Identification of the Key microRNAs and miRNA-mRNA Interaction Networks during the Ovarian Development of Hens

**DOI:** 10.3390/ani10091680

**Published:** 2020-09-17

**Authors:** Jing Li, Chong Li, Qi Li, Wen-Ting Li, Hong Li, Guo-Xi Li, Xiang-Tao Kang, Xiao-Jun Liu, Ya-Dong Tian

**Affiliations:** College of Animal Science and Technology, Henan Agricultural University, Zhengzhou 450046, China; lijingbj_87@126.com (J.L.); lc0924ll@163.com (C.L.); m18838939701@163.com (Q.L.); liwenting_5959@hotmail.com (W.-T.L.); lihong19871202@163.com (H.L.); liguoxi0914@126.com (G.-X.L.); xtkang2001@263.net (X.-T.K.); xjliu2008@hotmail.com (X.-J.L.)

**Keywords:** DE miRNAs, ovarian function, target genes, different stages, hens

## Abstract

**Simple Summary:**

In this study, we analyzed the microRNA (miRNA) sequencing libraries of the hen ovary at four different developmental stages (15, 20, 30, and 68 W) and found 209 differently expressed (DE) miRNAs in the six comparisons of the four stages, which might be involved in ovarian growth and functions. In particular, five key miRNAs (gga-miR-2954, gga-miR-6634-5p, gga-miR-449b-5p, gga-miR-449c-3p, and gga-miR449c-5p) targeting 23 predicted genes were obtained in the interaction network and functional enrichment analysis through a combination of DE miRNAs and differentially expressed genes (DEGs), which might be involved in cell differentiation and proliferation, steroid hormone biosynthesis, and angiogenesis in ovarian development. Our findings can contribute to a better understanding of the role of functional miRNAs in ovarian development and the ovulatory cycle during the four developmental stages of hens.

**Abstract:**

It is well-known that multiple functional miRNAs are found in mammals’ ovaries, which are linked not only to ovarian development, but also to maturation and apoptosis. However, there is still a lack of knowledge regarding the role of miRNAs in the hen ovary. In the present study, we analyzed the miRNA sequencing libraries of ovaries at the four different developmental stages of hens (15, 20, 30, and 68 W) and a total of 677 known miRNAs and 61 novel miRNAs were identified. In total, 209 of them were differently expressed miRNAs (DE miRNAs) obtained from comparisons of the four stages, including 84 upregulated and 125 downregulated DE miRNAs. Furthermore, the five key DE miRNAs gga-miR-2954, gga-miR-6634-5p, gga-miR-449b-5p, gga-miR-449c-3p, and gga-miR449c-5p were screened using an analysis of the miRNA-mRNA interaction network and functional enrichment annotated in seven significantly enriched pathways, such as endocytosis, lysine degradation, the biosynthesis of amino acids, and the MAPK signaling pathway, which may primarily participate in cell differentiation and proliferation, steroid hormone biosynthesis, and angiogenesis by targeting the related genes. For instance, gga-miR-449 family members were predicted to target 15 genes, including *TGFB1*, *TPM1*, *TPM3*, and *CAMKB2*, which were reported to regulate follicular growth, selection, and the ovulatory cycle. Taken together, our results illustrate the ovarian miRNA profiles of the four classic developmental stages of hens and highlight the significant role of miRNAs in ovarian development and functions. However, in-depth research needs to be carried out to validate the potential functional miRNAs found in this study.

## 1. Introduction

The ovarian status with age-alteration is directly associated with the laying mechanism of hens, which regularly triggers ovulation and maintains the laying process of chickens. Each ovarian follicle is comprised of a follicular theca layer, a granulosa layer, and stroma surrounding an oocyte, and the structure and numbers of follicular cells differ significantly in ovarian developmental stages [1]. It has been reported that there are numerous regulatory factors involved in the regulation of follicular development and ovarian functions in chickens, such as *IGFs*, *TGF-β*, *SF-1*, and *FOXL2*, which can regulate the expression of related genes in follicular selection, growth and maturation, cell differentiation, and steroid hormone biosynthesis through various signaling pathways [2,3,4,5].

MicroRNAs (miRNAs) are well-known to be short noncoding RNAs, around 21–25 nt in length, that are found widely in eukaryotic cells and negatively regulate the transcription of gene expression by binding the 3’UTR region of the target gene [6]. In the few last years, a large number of functional miRNAs in mammalian ovaries have been shown to play important roles in the regulation of follicular development and atresia, cell proliferation and apoptosis, steroid hormone biosynthesis, and ovarian diseases [7,8,9,10,11]. For example, miR126-3p can promote porcine follicular cell proliferation by regulating *PIK3R2* expression, and several miRNAs, including miR-26b, miR-34a, miR-125a, and miR-92a, can induce follicular granular cell apoptosis by targeting their individual genes [12,13]. Moreover, miR-375, miR-873, and miR-202 can influence the biosynthesis of steroid hormones in the ovary by inhibiting the expression of key genes [14,15]. In avian ovaries, miR-26a-5p was found to promote follicular theca cell proliferation, which was differently expressed in the mature and immature ovaries of chickens [16]. Additionally, miR-1b-3p was reported to possibly regulate the follicle development of hens [17]. However, research into functional miRNAs in the ovarian development and functions of chickens is still lagging behind and the process whereby the whole miRNA profiles in poultry ovaries are integrated during the reproductive process remains unclear. Therefore, the objective of our study was to explore the novel miRNAs that regulate follicular growth and the ovulatory cycle during the ages of 15, 20, 30, and 68 W, which represent the four critical stages of ovarian development: Initial development, first laying, and the high and low laying periods, respectively.

## 2. Materials and Methods

### 2.1. Ethics Statement

All experimental animals were treated while following the regulations for the administration of affairs concerning experimental animals. The animal care methods were approved by the Henan Agricultural University Institutional Animal Care and Use Committee (Permit Number: 19-0068).

### 2.2. Sample Collection, RNA Isolation, and Quality Analysis

A total of 36 Hy-line brown laying hens were collected at 15, 20, 30, and 68 weeks of age and were anesthetized by an intravenous wing injection of pentobarbital sodium (3%, 30 mg/kg body weight). Under deep anesthesia, each individual was euthanized by jugular vein bleeding. Then, we separated the whole ovarian tissues from all of the individuals, and froze and stored them at −80 ℃ in an ultra-freezer for the subsequent procedures. The total RNA of ovarian tissues was extracted using the Trizol RNA extraction reagent (Invitrogen, Carlsbad, CA, USA), and a total of 12 RNA samples was obtained, with each sample mixing three tissues at each age, in order to eliminate the differences between individuals. The purity and concentration of the RNA were determined using a NanoPhotometer^®^ spectrophotometer (IMPLEN, Westlake Village, CA, USA) and a Qubit^®^ RNA Assay Kit (Life Technologies, Carlsbad, CA, USA). The integrity of the RNA was evaluated using a Nano 6000 Assay Kit of the Bioanalyzer 2100 system (Agilent Technologies, Palo Alto, CA, USA).

### 2.3. Library Preparation and Sequencing

Three micrograms of total RNA per sample were prepared as the input for the small RNA library. The preparation and sequencing of the library were carried out using NEBNext^®^ Mutiple Small RNA library Prep Set (NEB, Lpswich, MA, USA) and the Illumina Hiseq 2500/2000 platform (Illumina, San Diego, CA, USA) to generate 50 bp single-end reads. The quality of the library was measured using an Agilent Bioanalyzer 2100 system (Agilent Technologies, Palo Alto, CA, USA) with DNA High Sensitivity Chips. A total of 12 small RNA libraries were obtained in this study.

### 2.4. Distribution and Identification of miRNA

Low quality reads, 3′ adapter/insert tag deletion, 5′ adapter contaminants, and Ploy A/T/G/C in the raw data were removed to obtain clean reads using Illumina CASAVA (version 1.8, http://www.illumina.com/) in the preprocessing. A length of 18–30 nt from the clean reads was filtered for the downstream analysis. Known miRNA alignments were identified using miRBase 20.0 and described those that started with “gga-miR-”. Novel miRNA was predicted using miREvo and mirdeep2 [18,19], describing that which started with “novel-”.

### 2.5. Differential Expression of miRNA and mRNA and Clustering Analysis

The miRNA and mRNA expression levels were presented as the transcript per million (TPM) and fragments per kilo-base of exon per million fragments mapped (FPKM), respectively [20,21]. The differential expression of miRNA and mRNA between every two ages was performed using the DESeq R package (version 3.4.2, http://www.R-project.org/). An adjusted *p* value < 0.05 and |log2 (fold change)| > 1 were set as the thresholds to identify significantly differently expressed miRNAs (DE miRNAs) and differently expressed mRNA (DE mRNA). Furthermore, a Venn diagram was produced using the limma package in R (version 3.32.5, http://www.bioconductor.org/packages/release/bioc/html/limma.html). Hierarchical clustering analyses were used to demonstrate the expression profiles of miRNAs during different ovarian development stages of hens.

### 2.6. Target Gene Prediction, Enrichment, and Interaction Network Analysis

The target genes of miRNAs were predicted by psRobot_tar in miRanda for animals and integrated with the differentially expressed gene (DEG) profiles to reinforce the accuracy of prediction [22]. In this study, the raw profile of mRNA was submitted in the NCBI database and contributed to analyzing the prediction of the target gene (more details in the Data Availability section). The negative correlation of miRNA and mRNA was selected by calculating the Pearson value. Cytoscape (version 3.7.2, http://www.cytoscape.org/) was used for the miRNA-mRNA interaction network analysis. All of the potential target genes of DE miRNAs among the combinations of the four developmental stages were used for gene ontology (GO) enrichment and Kyoto Encyclopedia of Genes and Genomes (KEGG) pathway analyses, which were implemented by the GOseq R package (version 2.12, http://www.geneontology.org/) and KOBAS software (version 2.0, http://www.genome.jp/kegg/). GO terms with a *p*-value < 0.01 and KEGG pathways with *p* < 0.05 were considered to be significantly enriched.

### 2.7. Validation of miRNA Expression

To validate the RNA-seq of the miRNA data, we analyzed five highly expressed miRNAs using reverse transcription real-time PCR (RT-qPCR). Bulge-loop miRNA RT-qPCR primer sets (RiboBio, Guangzhou, China) were used to amplify miRNAs and the process of RT-qPCR was performed, following the manufacturer’s recommendation. The results were analyzed using the 2^−∆∆Ct^ method. The primer sequences of the selected miRNAs and internal reference-chicken U6 RNA were designed via RiboBio (RiboBio, Guangzhou, China).

## 3. Results

### 3.1. Sequencing Analysis

In the sequencing libraries, we obtained 12,739,770 raw reads, 12,604,754 clean reads, and 11,760,062 mapped reads, with a 99.76% Q20 content on average, which indicated the high quality of the Illumina sequencing. In addition, the miRNA sequences of the ovaries in the four developmental stages were generally 21–23 nt in length, occupying 83.99% (Table 1). A total of 738 miRNAs were generated in our transcriptome data, including 677 known miRNAs and 61 novel miRNAs, which were used for the following analyses.

### 3.2. DE miRNA Analysis

To reveal the distribution of DE miRNAs, a bar graph, heatmap, and Venn diagram were developed to demonstrate the up- and downregulated miRNAs, the expression profiles, and the co-expressed/specific miRNAs among the combinations of different developmental stages. In total, 209 DE miRNAs were found in our data, with 84 upregulated and 125 downregulated DE miRNAs. In particular, the maximum number of DE miRNAs (69) was obtained in the comparison of O30 and O15, which indicated variation in the preliminary and vigorous growth stages of the hen ovary compared to the others (Figure 1A). Furthermore, there were similar expression models of miRNAs between O30 and O68, most of which were different from those in O15 and O30 (Figure 1B). In the Venn diagrams, no co-expressed miRNAs were found when considering all six combinations (Figure 1C). However, were eight co-expressed DE miRNAs (e.g., gga-miR-135a-5p, gga-miR-202-5p, gaa-miR-31-5p, gga-miR-34a-5p, gga-miR-449a, gga-miR-449b-5p, gga-miR-449d-5p, and gga-miR-499-5p) were obtained in the comparisons of O30 vs. O15, O68 vs. O15, O30 vs. O20, and O68 vs. O20, which might be involved in the physiological functions of the ovary before and after the egg-laying of hens.

### 3.3. Interaction Network Analysis of miRNA-mRNA

To investigate the potential interaction of miRNA-mRNA, a miRNA-mRNA network was constructed based on a comparison of the predicted target genes of all DE miRNAs and DEG profiles among the six combinations, as well as the negative correlation of DE miRNA-mRNA with the different expression models at the four ages of the hens. Within the interaction network, 51 miRNAs and 290 predicted genes formed 479 negative miRNA-mRNA regulatory networks, among which there were 86 upregulated miRNAs with downregulated genes and 393 downregulated miRNAs with upregulated genes (Figure 2A,B). The top five miRNAs associated with the largest number of target genes were gga-miR-449b-5p, gga-miR449c-5p, gga-miR-449c-3p, gga-miR-6634-5p, and gga-miR-2954, respectively. Notably, all of them were downregulated DE miRNAs during the different developmental stages. The concrete profile of the interaction network, the expression levels, and Pearson’s correlation analyses of DE miRNAs and DE mRNAs are listed in Appendix A.

### 3.4. Functional Enrichment Analysis

Using the predicted target genes of the DE miRNAs obtained from the interaction network analysis, we performed a GO and KEGG enrichment analysis to reveal the potential role of DE miRNAs in the ovarian function. The top 30 enriched GO terms are shown in Figure 3A, indicating that the most enriched BP, CC, and MF terms responded to the biotic stimulus, actin cytoskeleton, and vitamin binding, respectively. Moreover, the BP terms mainly focused on the responses to external stimuli, cell differentiation and proliferation, and the immune system, which are strongly associated with tissue development. Furthermore, it was revealed that 101 target genes were annotated in 92 pathways, among which seven significantly enriched pathways were obtained, including endocytosis, the intestinal immune network for IgA production, the biosynthesis of amino acids, lysine degradation, adrenergic signaling in cardiomyocytes, the MAPK signaling pathway, and fructose and mannose metabolism (*p* < 0.05) (Figure 3B). To further explore the key miRNAs and predicted target genes, we constructed a potential correlation network of the miRNAs-gene-pathway-function and demonstrated that gga-miR-499c-3p, gga-miR-499b-5p, gga-miR-499c-5p, gga-miR-2954, and gaa-miR-6634-5p were widely involved in the regulation of multiple target genes in various significantly enriched pathways (Figure 3C). In particular, the predicted target genes of gga-miR-449c-5p and gga-miR-2954–*TGFB1* were involved in both the endocytosis and MAPK signaling pathways, and *AADAT* was involved in both the biosynthesis of amino acids and lysine degradation, respectively. Interestingly, the five key miRNAs were the same as those obtained in the interaction network of miRNA-mRNA, which suggests that these miRNAs have play major roles in ovarian development and the ovulatory cycle, such as cell differentiation and proliferation, steroid hormone biosynthesis, and angiogenesis. The concrete profiles of GO and KEGG enrichment analysis are listed in Appendix A, respectively.

### 3.5. Integrated Analysis of Key DE miRNAs

According to the above analysis of the interaction network and enrichment annotation, we integrated the key possible miRNA-mRNA networks that may modulate the ovarian function during the different developmental stages. It was shown that the five key miRNAs associated with a total of 23 target genes formed 30 combined pairs in this network, among which gga-miR-449b-5p was co-expressed with DE miRNAs in the comparisons of O30 vs. O15, O30 vs. O20, O68 vs. O15, and O68 vs. O20, as previously mentioned, which might regulate the expression of *NFATC1*, *STAMBP, BF2,* and *OGDH*. Meanwhile, *OGDH* and *BF2* were both targets of gga-miR-449b-5p, gga-miT-449c-5p, and gga-miR-2954. *TPM3* was the target of gga-miR-449c-5p and gga-miR-6634-5p, and *BRT-1* was the target of gga-miR-2954 and gga-6634-5p. All of the results indicate that these key miRNAs might be involved in the physiological process of ovarian development and function through the related signaling pathways (Figure 4).

### 3.6. Validation of miRNA Profiles by qRT-PCR

Using these data, we validated six highly expressed miRNAs (e.g., gga-miR-103-3p, gga-miR-7b, gga-miR-21-5p, gga-miR-24-3p, gga-miR-31-5p, and gga-miR-146b-5p) using qRT-PCR (Figure 5). All of the expression results are consistent with those in RNA-seq data and provided a great reliability for conducting the subsequent research.

## 4. Discussion

The ovarian morphology and functions were extremely variable in the different developmental stages of the hens. The ovary developed very slowly, with several primary and secondary follicles appearing before sexual maturation, while the rapid growth of the ovary mainly occurred after laying the first egg [23]. The ovarian shape during the peak laying period was much larger than that in the other periods and consisted of numerous follicles with various sizes to maintain the process of frequent egg laying [24]. Then, the capacity of egg production declined with age due to a high rate of follicle atresia in the ovary [25]. Based on this result, we sought to assess whether functional miRNAs play a crucial role in ovarian development and functions along with the changing age of hens. Therefore, in this study, we determined the miRNA profiles in the four classic stages of ovarian development (15, 20, 30, and 68 W) using RNA-seq, which also represented the first study analyzing all four different stages of hens’ ovarian development. A total of 677 known miRNAs and 61 novel miRNAs were identified, and 209 DE miRNAs were obtained among the comparisons of the four stages of development. Throughout all the DE miRNAs, there were 84 upregulated and 125 downregulated DE miRNAs in our data, of which eight co-existing DE miRNAs (gga-miR-135a-5p, gga-miR-202-5p, gaa-miR-31-5p, gga-miR-34a-5p, and gga-miR-449 family members) were observed in the comparisons of O30 vs. O15, O30 vs. O15, O30 vs. O20, O68 vs. O15, and O68 vs. O15 using Venn diagrams, likely related to the ovarian maturation and ovulation mechanism. Furthermore, miR-202-5p, miR-34a, miR-31-5p, and miR-135a have been reported to be involved in ovarian granulosa cellular proliferation, apoptosis, and steroidogenesis in mammals [7,26,27,28]. However, the role of mi-449 family members in the ovary remains unclear, except for miR-499a and miR-449b, which have been proven to lead to cell cycle arrest in ovarian cancer cells [29].

In the GO and KEGG enrichment analyses, we found that the BP categories of the top 30 GO terms and the significantly enriched pathways were both involved in the regulation of tissue development and the immune system, which responded to the ovarian alteration status with age. To more deeply excavate the key miRNAs and target genes, we developed an interaction network of miRNA-mRNA and enrichment analyses of the target genes, indicating that gga-miR-2954, gga-miR-6634-5p, gga-miR-449b-5p, gga-miR-449c-3p, and gga-miR-449c-5p did not only widely target multiple genes, but were also significantly enriched in seven pathways, including the endocytosis, biosynthesis of amino acids, and MAPK signaling pathways (*p* < 0.05). In addition, all of these genes were downregulated by differently expressed miRNAs.

Then, we filtered 23 target genes in the binding of the five key miRNAs that may be associated with ovarian development and function and obtained a total of 30 combined pairs. Some studies have reported gga-miR-2954 to be an essential miRNA related to the sex differences of poultry [30,31]. However, our results predict that it is an important DE miRNA among the developmental stages of hens, which was also determined in the granulosa and thecal layer of geese ovarian follicles, indicating that gga-miR-295 may be related to follicle selection [32]. Unfortunately, there was no evidence demonstrating the role of gga-miR-6634-5p in the ovarian development and function of hens. Furthermore, it was interesting that the gga-miR-449 family members co-existed in the comparison of before (15 and 20 W) and after (30 and 68 W) the continuous laying periods and were involved in all seven significantly enriched pathways by targeting 15 genes, among which *TGFB1* was linked to endocytosis and the MAPK signaling pathway that play a major role in steroid hormones synthesis and follicular development [33,34]; *TPM1* was involved in the regulation of angiogenesis by reducing the *VEGF* expression [35]; and *TPM3*, *ATP2B4*, *CAMK2B,* and *NFATC1* might also be associated with angiogenesis. Clearly, both steroidogenesis and angiogenesis in follicles significantly differed and played a crucial role in follicular selection, maturation, and the ovulatory cycle during the ovarian developmental stages of hens, in which the miR-449 family may contribute to regulation by targeting the related genes [36,37]. The miR-449 family has been suggested to be an indicator of a weak expression in cancer cells, inhibiting cell proliferation and migration to suppress tumor development in various tissues of mammals, such as the liver, stomach, and ovary [38,39,40]. However, the function of miR-449 in the normal ovaries of poultry is still unknown. Therefore, further research must be performed to explore the roles of key miRNAs in ovarian functions.

## 5. Conclusions

In summary, our study highlighted the role of miRNAs in ovarian development and functions, first outlining the transcriptome analysis of miRNAs during the four classic ovarian developmental stages of hens. Meanwhile, it was indicated that five key DE miRNAs, including gga-miR-449b-5p, gga-miR-499c-3p, gga-miR-449c-5p, gga-miR-2954, and gga-miR-6634-5p, might play a major role in the process of follicular development, steroidogenesis, and angiogenesis. Therefore, our findings will contribute to a better understanding of the role of miRNAs in ovarian development and physiological functions during the four stages of the hen ovary.

## Figures and Tables

**Figure 1 animals-10-01680-f001:**
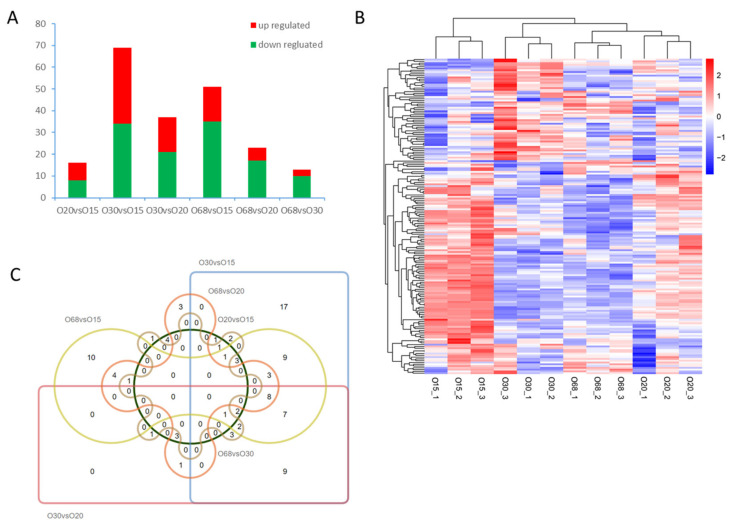
The distribution of differentially expressed microRNAs (DE miRNAs) in different developmental stages. (**A**) The numbers of upregulated and downregulated DE miRNAs among the six comparison groups. (**B**) Hierarchical clustering heatmap of the DE miRNA expression profiles in the four developmental stages. (**C**) Venn diagrams of the DE miRNAs among the six comparison groups.

**Figure 2 animals-10-01680-f002:**
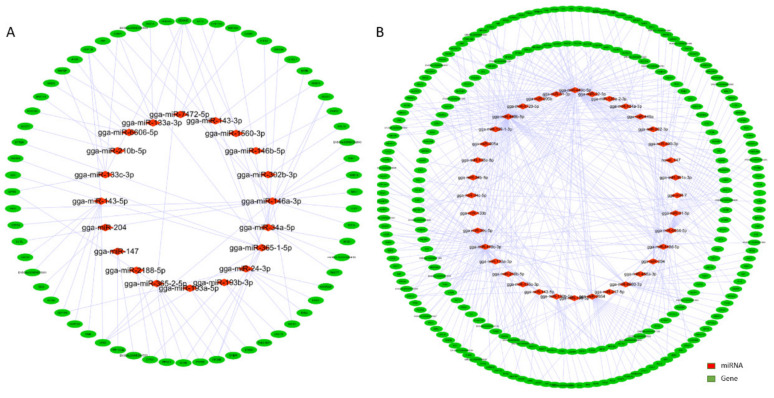
The interaction network of DE miRNA-differentially expressed gene (DEG). (**A**) The interaction network of the upregulated miRNA and downregulated target genes. (**B**) The interaction network of downregulated miRNAs and upregulated target genes. The red rhombus and green oval represent the miRNAs and predicted target genes, respectively.

**Figure 3 animals-10-01680-f003:**
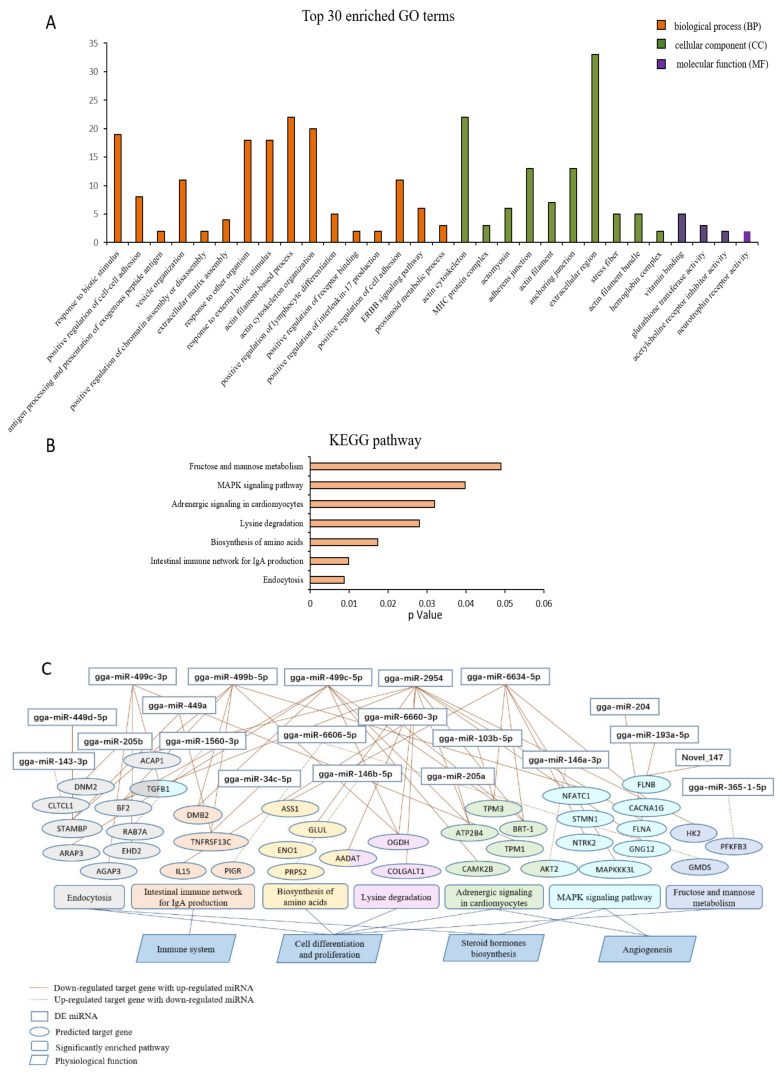
Functional enrichment analysis for DE miRNAs-DEGs. (**A**) The top 30 gene ontology (GO) terms from the GO enrichment analysis of the predicted target genes. (**B**) The Kyoto Encyclopedia of Genes and Genomes (KEGG) pathway significant enrichment analysis of predicted target genes. (**C**) The potential correlation network of miRNA-gene-pathway-function.

**Figure 4 animals-10-01680-f004:**
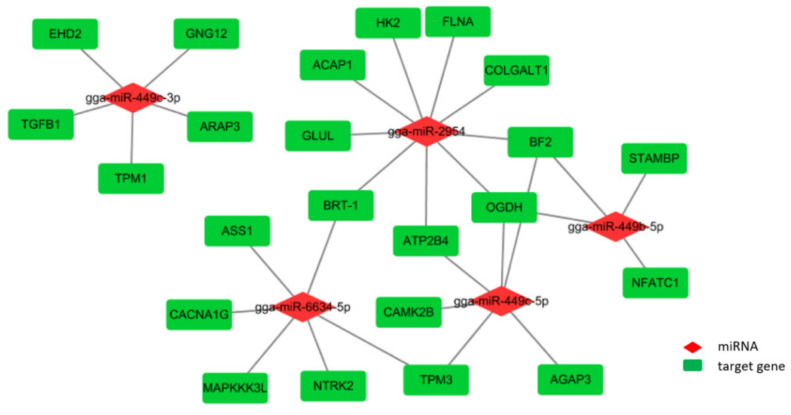
Integrated key miRNA-mRNA network. The red rhombus and green rectangle represent the key miRNAs and predicted target genes, respectively.

**Figure 5 animals-10-01680-f005:**
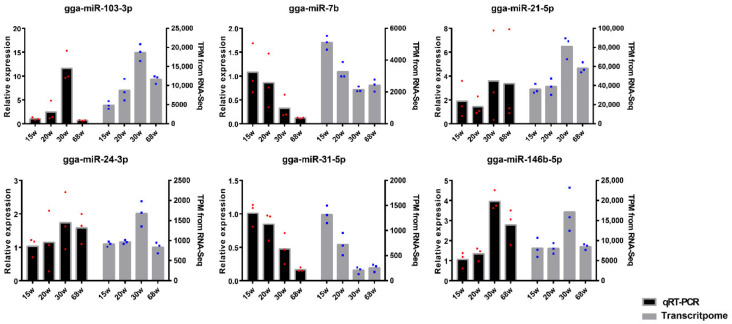
The validation of highly expressed miRNAs between qRT-PCR and the transcriptome. The left y-axis of the above graph shows the value of relative expression and the right y-axis shows the value of transcript per million (TPM). The red and blue dots correspond to the three biological replicates.

**Table 1 animals-10-01680-t001:** Overview of sequencing data for small RNA.

Samples	Raw Reads	Clean Reads	Clean Ratio (%)	Mapped Reads	Mapped Ratio (%)	Q20 (%)	Frequency Percent of 21–23 nt (%)
O15_1	12,668,758	12,485,845	98.56	9,568,340	94.1	99.81	79.25
O15_2	17,026,099	16,850,839	98.97	11,052,700	96.31	99.76	80.54
O15_3	11,521,207	11,369,282	98.68	11,614,036	95.84	99.69	79.78
O20_1	13,211,633	13,123,238	99.33	10,360,873	94.65	99.69	77.09
O20_2	13,225,084	13,073,265	98.85	10,436,434	95.82	99.73	86.56
O20_3	11,191,768	11,061,677	98.84	10,414,140	96.03	99.7	85.07
O30_1	12,553,362	12,421,066	98.95	10,578,899	95.37	99.74	83.34
O30_2	11,766,007	11,652,830	99.04	11,540,385	95.37	99.81	87.53
O30_3	11,300,865	11,136,431	98.54	12,259,060	96.56	99.7	87.78
O68_1	16,894,428	16,730,280	99.03	12,135,277	96.15	99.67	85.81
O68_2	11,172,817	11,062,162	99.01	15,619,129	95.14	99.82	87.39
O68_3	10,345,214	10,290,138	99.47	15,541,471	94.27	99.85	87.72
Average	12,739,770	12,604,754	98.94	11,760,062	95.47	99.76	83.99

Note: Each sample name consists of letters and numerals. The capital letters with numerals 15, 20, 30, and 68 correspond to the tissues of the ovary and the four different weeks of developmental stages, while the numerals 1, 2, and 3 correspond to the three samples in each stage. This naming convention is the same as that used in the following tables and figures.

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
