# Peer review of "Identification of the Key microRNAs and miRNA-mRNA Interaction Networks during the Ovarian Development of Hens"

_animals, 2020, doi:10.3390/ani10091680_

Round 1

Reviewer 1 Report

I have found some minor mistakes and need to be clarify by first authors.

(1) Line 15-16 & 29-30: Authors kept mentioning 5 key miRNAs, however, only 4 genes listed. If this is an counting mistake, change 5 key miRNAs to 4 or add the name of a missing miRNA to the list.

(2) Supplement table S1 & S2 are missing for the reviewer or reader.

(3) Line 175: In figure 3A only 29 enriched GO terms were shown not 30 which authors had claimed.

Author Response

Dear Reviwer:

  Many thanks for your great comments, I have modified in my manuscript and the feedback as follows:

(1) Line 15-16 & 29-30: Authors kept mentioning 5 key miRNAs, however, only 4 genes listed. If this is an counting mistake, change 5 key miRNAs to 4 or add the name of a missing miRNA to the list.

Answer: Thank you very much for the reminding. Actually there are 5 key miRNAs in our finding, and I typed gga-miR-449c-3p and gga-miR-449c-5p together to gga-miR-449c-3p/5p, so sorry for the inappropriate typing and mistake. Now we have separated them in the manuscript.

(2) Supplement table S1 & S2 are missing for the reviewer or reader.

Answer: Thank you very much for the reminding. We have realized that there were something wrong with the supplementary documents and the related description in the manuscript. We have corrected the supplement tables (e.g. table S1, table S2 and table 3), which were consistent with the description of the manuscript.

(3) Line 175: In figure 3A only 29 enriched GO terms were shown not 30 which authors had claimed.

Answer: Thank you very much for the reminding. There was a mistake to upload the figure 3A due to we have changed it. And this mistake has been corrected in the manuscript.

Very appreciate for your great comments.

Reviewer 2 Report

It would be an interesting addition about miRNAs regulation of ovarian development and function in hens. There are still some grammatical errors. In the introduction, the authors need to briefly justify that why 15W, 20W, 30W and 68W of ages have been selected. Rather than saying as four critical stages, the authors need to correlate the physiology of ovarian development to the chosen ages (15W, 20W, 30W and 68W). Differential expression of genes (mRNAs) need to be included in the manuscript since correlation between differential expression of miRNAs and mRNAs (genes) are being discussed. Otherwise, the authors can completely avoid discussing the correlation (direct or indirect: in most cases there will be inverse correlations) between DE of miRNAs and DEG and can go with the integration of genes and pathways. Overall, it would be a good addition to the literature in specific to miRNAs regulation of ovarian development and function in hens if these corrections have been made.   

Author Response

Dear Reviewer:

   Thank you for your great comments, I have modified all the comments in my manuscript, the feedback is as follows:

 It would be an interesting addition about miRNAs regulation of ovarian development and function in hens. There are still some grammatical errors. In the introduction, the authors need to briefly justify that why 15W, 20W, 30W and 68W of ages have been selected. Rather than saying as four critical stages, the authors need to correlate the physiology of ovarian development to the chosen ages (15W, 20W, 30W and 68W). Differential expression of genes (mRNAs) need to be included in the manuscript since correlation between differential expression of miRNAs and mRNAs (genes) are being discussed. Otherwise, the authors can completely avoid discussing the correlation (direct or indirect: in most cases there will be inverse correlations) between DE of miRNAs and DEG and can go with the integration of genes and pathways. Overall, it would be a good addition to the literature in specific to miRNAs regulation of ovarian development and function in hens if these corrections have been made.

Answer: Thank you very much for your great suggestion. And we have revised the manuscript according to the comments, the major changes are as follows:

1) in the introduction, we highlighted the four ages and modified the final sentence ‘However, the research of functional miRNA in ovarian development and function of chicken is still relatively lagging behind and the analysis of integrated entire miRNA profiles in poultry ovary during the whole reproductive process is unclear. Therefore, the objective of our study was an attempt to explore the novel miRNAs that regulate follicular growth and ovulatory circle during the ages of 15w, 20w, 30w, 68w, which represent the four critical stages of ovarian development–initial development, first laying, high and low laying periods. ’ (line 63-69)

2)For the profile of differential expression of genes (mRNAs), we didn’t mention the more details of mRNAs in this manuscript due to the profile of mRNAs have been used in another manuscript. In addition, we focused on the miRNA profile and DE miRNA-DEG correlation analysis to find the key miRNAs that might be involved in ovarian and development and function through regulating the target genes, and we also noted in line 110-112, which was marked in red. If we only analysis the miRNA profile and predict target genes without linking to the DEGs, there would be a huge number of the predicted target genes, it’s hard to find the valuable miRNAs.

3)We have checked the grammar of the whole manuscript and corrected the grammatical mistake.  

Round 2

Reviewer 2 Report

Rereview Comments:

Although authors addressed one of the comments (justifying the physiological significance of ages of hens that have been selected in the objectives of the study) they failed to address the other two important concerns.

Still the manuscripts have some minor grammatical error. For example, on line 39, "valid" should be a verb.

If the manuscripts shows a negative correlation between miRNAs and mRNAs, the manuscript should include mRNA profile, at least corresponding mRNAs for the top group of differentially expressed miRNAs. It will be a difficult process. First one needs to bring up the significant differentially expressed miRNAs between groups. Then one needs to pick up regulated mRNAs by these already selected miRNAs for the same comparison group. Subsequently, one can identify the correlation. Please note that this is one way of looking at the correlation/relationship between miRNAs and mRNAs. In the manuscript, it is stated that mRNA profile is in the NCBI database whereas in addressing reviewer's comments, it is stated as mRNA profile is in another publication. Why are these information contradicted? The manuscript doesn't need mRNA profile to integrate miRNAs and target genes if the manuscript does not state about negative correlation. Top statistically significant and differentially expressed miRNAs can be picked up and integrated to target genes to predict biological pathways.  

Author Response

Dear Reviewer,

    Thank you very much for your great suggestions. And so sorry for the misunderstanding of the issue of the mRNA profiles in our manuscript. You’re right we should provide the DE mRNA profiles, at least the information of DE mRNA related to DE miRNA. It’s necessary to do the following analyses. Therefore, we have provided the interaction network, the expression levels (at the four ages of hens) and Pearson’s correlation coefficient of DE miRNA and DE mRNA in the Supplement Table S1 in our manuscript. Then we selected all the negative correlation of DE miRNA-DE mRNA according to the above analysis and constructed the interaction network using PPI and found the key miRNAs targeted gene, which may be involved in the ovarian development and functions. Furthermore, we have amended the description of 2.6 and 2.7 in the method part and 3.3 in the result part.  Please find more details in the revised manuscript. 

In addition, for the grammatical error, we re-checked the whole manuscript and corrected the grammatical and English-editing mistake. Thanks again for your great comments.

Best Regards,

Yadong Tian and Jing Li

Round 3

Reviewer 2 Report

This revised manuscript still needs grammar corrections. Some but not all are

Line 16: “gene” needs to be replaced by “genes”

Line 33: “may primarily participated” should be replaced by “may primarily participate”

Line 37: Remove the hyphen

Line 38: “in-depth” is better word for “deep”

Line 78: Change “fridge” to “ultra-freezer”. Fridge’s (refrigerator) temperature is 4°C

Line 79: “subsequent procedures” would be better in place of “following research”

Line 109: Replace “target gene of miRNA” by “target genes of miRNAs”

Line 111: Replace “were submitted in” by “was submitted to”

Line 122: “amplify” would be better for “extract”

Line 157: Change “O30vs015” to “O30vsO15”

Author Response

Dear reviewer,

    I'm very appreciate for your great comments, which are very helpful to make this manuscript much better. Thank you so much. And the grammar corrections are as follows:  

Line 16: “gene” needs to be replaced by “genes”

Answer: Thank you very much for your great suggestion. We have corrected in the revised manuscript, please check it.

Line 33: “may primarily participated” should be replaced by “may primarily participate”

Answer: Thank you very much for your great suggestion. We have corrected in the revised manuscript, please check it.

Line 37: Remove the hyphen

Answer: Thank you very much for your great suggestion. We have corrected in the revised manuscript, please check it.

Line 38: “in-depth” is better word for “deep”

Answer: Thank you very much for your great suggestion. We have corrected in the revised manuscript, please check it.

Line 78: Change “fridge” to “ultra-freezer”. Fridge’s (refrigerator) temperature is 4°C

Answer: Thank you very much for your great suggestion. We have corrected in the revised manuscript, please check it.

Line 79: “subsequent procedures” would be better in place of “following research”

Answer: Thank you very much for your great suggestion. We have corrected in the revised manuscript, please check it.

Line 109: Replace “target gene of miRNA” by “target genes of miRNAs”

Answer: Thank you very much for your great suggestion. We have corrected in the revised manuscript, please check it.

Line 111: Replace “were submitted in” by “was submitted to”

Answer: Thank you very much for your great suggestion. We have corrected in the revised manuscript, please check it.

Line 122: “amplify” would be better for “extract”

Answer: Thank you very much for your great suggestion. We have corrected in the revised manuscript, please check it.

Line 157: Change “O30vs015” to “O30vsO15”

Answer: Thank you very much for your great suggestion. We have corrected in the revised manuscript, please check it.